# Generative Multimodal Data Augmentation for Low-Resource Multimodal Named Entity Recognition

## ABSTRACT

As an important task in multimodal information extraction, Multimodal Named Entity Recognition (MNER) has recently attracted considerable attention. One key challenge of MNER lies in the lack of sufficient fine-grained annotated data, especially in low-resource scenarios. Although data augmentation is a widely used technique to tackle the above issue, it is challenging to simultaneously generate synthetic text-image pairs and their corresponding high-quality entity annotations. In this work, we propose a novel Generative Multimodal Data Augmentation (GMDA) framework for MNER, which contains two stages: Multimodal Text Generation and Multimodal Image Generation. Specifically, we first transform each annotated sentence into a linearized labeled sequence, and then train a Label-aware Multimodal Large Language Model (LMLLM) to generate the labeled sequence based on a label-aware prompt and its associated image. After using the trained LMLLM to generate synthetic labeled sentences, we further employ a Stable Diffusion model to generate the synthetic images that are semantically related to these sentences. Experimental results on three benchmark datasets demonstrate the effectiveness of the proposed GMDA framework, which consistently boosts the performance of several competitive methods for two subtasks of MNER in both full-supervision and low-resource settings.

## CCS CONCEPTS

• **Computing methodologies → Information extraction**; • **Information systems → Multimedia and multimodal retrieval**.

## KEYWORDS

multimodal named entity recognition, grounded multimodal named entity recognition, data augmentation, generative framework

## 1 INTRODUCTION

Recent years have witnessed an exponential growth of multimodal user posts on various social media platforms such as Twitter, Facebook, and Instagram. As a large amount of multimodal content often contains much important structured information such as named entities and their relations that are crucial for multimodal knowledge graph construction, multimodal information extraction has attracted increasing attention in recent years [23, 25, 43]. As

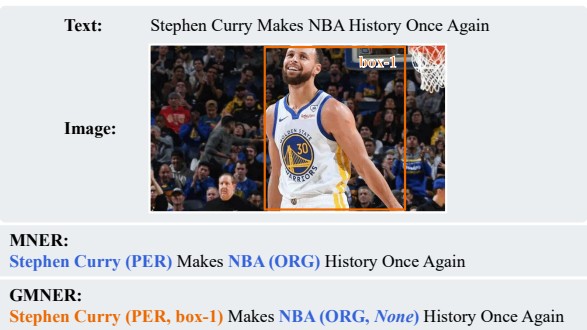

**Text:** Stephen Curry Makes NBA History Once Again

**Image:**

**MNER:**
**Stephen Curry (PER)** Makes **NBA (ORG)** History Once Again

**GMNER:**
**Stephen Curry (PER, box-1)** Makes **NBA (ORG, *None*)** History Once Again

**Figure 1:** An annotation example of Multimodal Named Entity Recognition (MNER) and Grounded Multimodal Named Entity Recognition (GMNER).

a fundamental task in multimodal information extraction, Multimodal Named Entity Recognition (MNER) aims to extract the named entities mentioned in an image-text pair and classify them into pre-defined types, such as person (PER), location (LOC), and organization (ORG) [41]. For example, given the multimodal tweet in Figure 1, an MNER system is expected to extract two entities, i.e., *Stephen Curry* and *NBA*, and their corresponding entity types are *PER* and *ORG*, respectively.

Existing approaches on the MNER task primarily include sequence labeling-based methods [4, 32, 38, 40], index generation-based methods [39], paraphrase generation-based methods [30], and in-context learning-based methods [3, 5]. Due to the emerging demand for multimodal knowledge graph construction, Yu et al. [39] recently introduced an extension task of MNER named Grounded MNER (GMNER). As shown in Figure 1, the goal of GMNER is to extract named entities, entity types and the bounding boxes of their grounded visual objects from text-image pairs. To address the GMNER task, existing studies mainly focus on either using a pipeline approach to decompose the task into several subtasks and solve them one by one [22, 26] or proposing an end-to-end approach to directly generate the entity-type-object triplets [30, 39].

One key challenge of the aforementioned methods is their heavy reliance on annotated data. As illustrated in Figure 1, both MNER and GMNER tasks require fine-grained annotation of textual named entities and their entity types, while GMNER further requires annotating the bounding box of visual objects that are corresponding to the named entities. In real applications, it is often time-consuming and costly to obtain such human annotation, which hinders the effectiveness of existing MNER and GMNER models in many low-resource scenarios.

One attractive solution to address the data sparsity issue is to automatically generate annotated data by data augmentation (DA). Existing DA methods for NER can be summarized into two groups: 1) using rule-based methods such as word replacement, shuffling, and cropping to obtain similar sentences [11, 29]; 2) using generation-based methods to directly generate the labeled sentences [12, 44].

However, all these DA methods solely focus on the textual modality, which cannot be directly applied to generate labeled text-image pairs. Compared with these text DA methods for NER, data augmentation for MNER and GMNER is more challenging for several reasons. First, it is necessary to generate both text and images, and each text-image pair should be semantically related. Second, each generated text-image pair is required to have the textual and visual entity annotations.

To address these challenges, in this paper, we propose a two-stage Generative Multimodal Data Augmentation framework for MNER named GMDA, which contains a Multimodal Text Generation stage to produce synthetic sentences with labeled entities and a Multimodal Image Generation stage to generate the corresponding image for each synthetic labeled sentence. Specifically, given a training sample, we first transform the input text and its entity labels into a linearized sentence, and then devise a Label-aware Multimodal Large Language Model (LMLLM) based on one of the representative MLLMs InstructBLIP [10], which is trained to generate the linearized sentence based on the input image and an entity label-aware instruction. During inference, for each training sample, we feed the input image and its entity label-aware instruction to the trained LMLLM and use a probability-based sampling strategy to generate a synthetic labeled sentence in an autoregressive manner. Based on the synthetic labeled sentence, the Multimodal Image Generation stage further employs a widely used latent diffusion model named Stable Diffusion [28] to generate a corresponding synthetic image conditioning on the synthetic sentence-based prompt and the original image.

The main contributions of our work can be summarized as follows:

- We propose a novel Generative Multimodal Data Augmentation framework named GMDA, which can generate a large number of text-image pairs with fine-grained entity annotations for both MNER and GMNER tasks.
- Under the GMDA framework, we devise a Label-aware Multimodal Large Language Model (LMLLM) to generate synthetic labeled sentences, followed by employing a latent diffusion model to generate the synthetic image for each labeled sentence.
- Extensive experiments on both MNER and GMNER tasks show that the proposed GMDA framework consistently boosts the performance of several competitive methods in both full-supervision and low-resource settings.

## 2 RELATED WORK

### 2.1 Multimodal Named Entity Recognition

Multimodal Named Entity Recognition (MNER) aims to recognize named entities in text and classify them into predefined categories based on text-image pairs. Pioneering works [23, 25, 41] focus on fusing visual information for improved word representation learning. With the use of the multimodal transformer architecture, a variety of attention-based mechanisms [1, 6, 7, 35–38] are designed to model the interactions between textual and visual modalities. In addition, converting images into natural languages [32] and retrieving external knowledge [21, 31] are used to enhance the textual information. Different from sequence labeling-based methods

mentioned above, machine reading comprehension (MRC)-based methods [2, 19], in-context learning-based methods [3, 5], index generation- based methods [39], and paraphrase generation-based methods [30] have been recently adapted to the MNER task.

Due to the emerging demand for multimodal knowledge graph construction, Grounded MNER (GMNER) is introduced as an extension of MNER, which aims to extract named entities, entity types and the bounding boxes of their grounded visual objects from text-image pairs. Existing end-to-end approaches [30, 39] formulate the GMNER task as multimodal index generation [39] and paraphrase and visual object generation [30], while pipeline approaches decompose GMNER into MNER, visual entailment and visual grounding task to solve one by one [22]. However, all these methods above heavily rely on a large amount of annotated data, which requires the fine-grained annotation of named entities and their entity types for MNER and the annotation of the bounding box of visual objects that are corresponding to the named entities for GMNER. Since obtaining such human annotation is time-consuming and costly, this work aims to propose an effective data augmentation method for MNER to enrich the annotated text-image pairs, particularly in low-resource settings.

### 2.2 Data Augmentation

Data Augmentation (DA) aims to increase the training data via slight changes of existing training data [34], which is widely used in various NLP tasks, especially in low-resource scenarios. In the literature, rule-based techniques were commonly employed, and typical approaches include word and mention replacement, segments shuffling [11], cropping, span rotation [29], random deletion [34], and subject/object inversion [24]. However, slight alterations in words may potentially disrupt the fluency of the sentence or compromise the coherent interpretation of annotation tags associated with labeled words. Thus, in recent years, many generation approaches [12, 44] have been proposed to enhance the diversity and preserving the label integrity in sentences.

For image data augmentation, Copy-Paste [15] is a simple but useful data augmentation method in object-aware tasks. For vision-language representation, MixGen [16] linearly interpolates images and concatenates text sequences to generate a new training sample. Nonetheless, data augmentation in MNER is more challenging due to the necessity of maintaining the semantic relatedness between image-text pairs and generating fine-grained entity annotations in both text and images. Therefore, this work aims to propose a new multimodal data augmentation method to generate paired multimodal synthetic data to address these challenges.

## 3 METHODOLOGY

In this section, we first introduce the task formulation and the overview of the proposed Generative Multimodal Data Augmentation (GMDA) framework, and then present the details of each module of GMDA.

### 3.1 Task Formulation

Given a sentence with $n$ words $s = (w_1, \ldots, w_n)$ and an associated image $v$ as the input, the goal of the MNER task is to extract a set

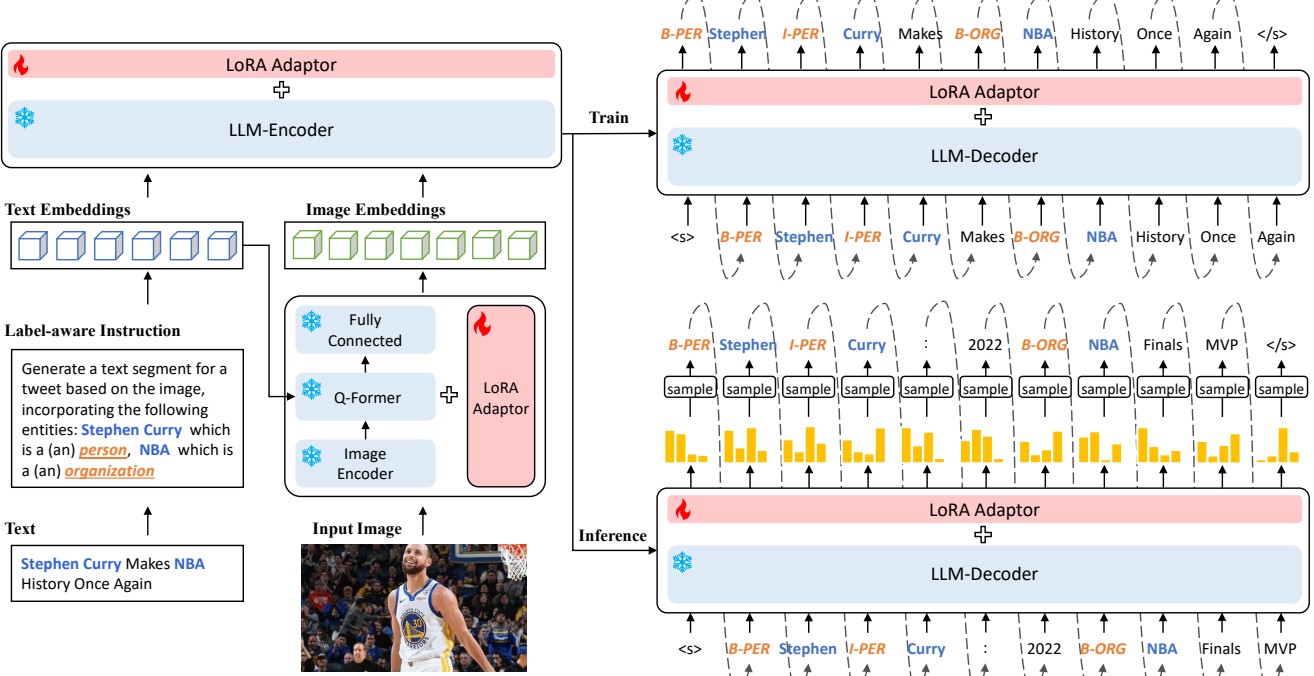

**Figure 2:** Overview of the proposed Label-aware Multimodal Language Model that generates synthetic labeled sentences in the Multimodal Text Generation stage.

of multimodal entity tuples:

$$\boldsymbol{y} = \{(e_1, t_1), \ldots, (e_m, t_m)\}, \quad (1)$$

and the goal of the GMNER task is to further extract the bounding boxes of the corresponding visual objects for each entity:

$$\boldsymbol{y} = \{(e_1, t_1, r_1), \ldots, (e_m, t_m, r_m)\}, \quad (2)$$

where $e_i$, $t_i$ and $r_i$ refer to the text span, the entity type, and the bounding box of the $i$-th entity in the input sentence $\boldsymbol{s}$. In both MNER and GMNER tasks, $t_i$ is one of the four pre-defined entity types, i.e., Person (PER), Location (LOC), Organization (ORG), and Miscellaneous (MISC). For the $i$-th entity, if there is no corresponding bounding box in the image, $r_i$ is *None*. Otherwise, $r_i$ consists of a 4-D spatial feature containing the top-left and bottom-right positions of the grounded bounding box, i.e., $(r_i^{x_1}, r_i^{y_1}, r_i^{x_2}, r_i^{y_2})$.

In this work, we mainly focus on a low-resource setting, in which there is a small set of labeled training data. Let $\mathcal{D} = \left\{ (\boldsymbol{s}_i, \boldsymbol{v}_i, \boldsymbol{y}_i) \right\}_{i=1}^N$ denote the training set. Our goal is to leverage $\mathcal{D}$ to generate another set of synthetic data $\mathcal{D}_g = \left\{ \left( \boldsymbol{s}_i^g, \boldsymbol{v}_i^g, \boldsymbol{y}_i^g \right) \right\}_{i=1}^K$.

### 3.2 Overview

As mentioned before, the proposed GMDA framework consists of two stages, i.e., Multimodal Text Generation and Multimodal Image Generation. As shown in Figure 2, in the first stage, we linearize the labeled sentence in each sample into a natural sequence, and then train a Label-aware Multimodal Large Language Model (LMLLM), which decodes the linearized sequence based on the image and the entity label of each sample as the inputs of the encoder. With the trained LMLLM, a probability-based sampling strategy is employed to generate a synthetic labeled sentence for each training sample

in an autoregressive manner. As shown in Figure 3, in the second stage, a widely used latent diffusion model is employed to generate a corresponding synthetic image conditioning on the synthetic sentence-based prompt and the original image.

### 3.3 Multimodal Text Generation

Given a sample $(\boldsymbol{s}_i, \boldsymbol{v}_i, \boldsymbol{y}_i)$ in the training set $\mathcal{D}$, the goal of this stage is to generate a new sentence $\boldsymbol{s}_i^g$ together with its entity labels $\boldsymbol{y}_i^g$. To ensure that the generated sentence is relevant to the original image and the generated entity labels have a high quality, we propose an encoder-decoder based conditional generation technique, which generates the linearized labeled sentence from the original image $\boldsymbol{v}_i$ and the original label $\boldsymbol{y}_i$.

*3.3.1 Linearized Labeled Sentences.* Firstly, we perform sentence linearization [12] by converting the labeled sentence into a natural sequence, in which the token labels are inserted before their corresponding words. For example, in Figure 2, the BIO tags of *Stephen*, *Curry* and *NBA* (i.e., *B-PER*, *I-PER*, and *B-ORG*) are inserted before each word. Note that since the tag *O* frequently occurs, we remove it to keep the linearized sentence more fluent.

*3.3.2 Label-aware Multimodal Large Language Model.* To generate the linearized sentence conditioning on the image and the entity labels, we propose a Label-aware Multimodal Large Language Model (LMLLM) based on a widely used MLLM named InstructBLIP [10].

**Label-Aware Instruction.** To guide the MLLM to better generate the linearized sentence, we first design an entity label-aware instruction, which contains the task description and the entity labels as "*Generate a text segment for a tweet based on the image, incorporating the following entities: $e_j$ which is a(n) $t_j$*", where $e_j$

and $t_j$ denote the $j$-th entity and its type, respectively. For example, in Figure 2, we linearize all labeled entities into a natural language sentence "*Stephen Curry which is a(an) person, NBA which is a(an) organization*". This implies that the entities are *Stephen Curry* and *NBA*, with their entity types being *PER* and *ORG*, respectively.

**Image Encoder.** To enable the image encoder of the pre-trained InstructBLIP model to acquire task-specific knowledge, we integrate LoRA adapter layers [18] for parameter-efficient fine-tuning. Specifically, given an input image $v$, we feed it to the image encoder with LoRA adapters to obtain the image representation $\mathbf{H}_v$:

$$\mathbf{H}_v = \text{Image-Encoder}(v; \theta_{\text{LoRA}}^{\text{img}}), \quad (3)$$

where $\theta_{\text{LoRA}}^{\text{img}}$ is the set of parameters to learn in LoRA.

**Q-Former.** The lightweight Querying Transformer (Q-Former) consists of an image transformer and a text transformer as submodules that share the same self-attention layers. LoRA adapter layers are also added to the attention layers as follows:

$$\mathbf{E}_v = \text{Q-Former}(\text{concat}(\mathbf{Q}, \mathbf{E}_P), \mathbf{H}_v; \theta_{\text{LoRA}}^{\text{Q-Former}}) \quad (4)$$

where $\mathbf{Q}$ and $\mathbf{E}_P$ refer to the learned queries and text embedding of label-aware instruction. In cross attention layers, $\mathbf{H}_v$ is regarded as the key and value while $\text{concat}(\mathbf{Q}, \mathbf{E}_P)$ is regarded as the query.

**LLM Encoder.** In contrast to InstructBLIP which solely fine-tunes the Q-former to align textual and visual features, our goal is to fine-tune the LLM to generate the linearized labeled sentence. Thus, we incorporate LoRA layers into the LLM for parameter-efficient fine-tuning. Specifically, the textual embeddings $\mathbf{E}_p$ of the label-aware instruction and the visual embeddings $\mathbf{E}_v$ are concatenated together and then fed into the LoRA-based LLM encoder to derive the hidden representation of the multimodal input:

$$\mathbf{H}_e = \text{LLM-Encoder}\left(\text{concat}(\mathbf{E}_p, \mathbf{E}_v); \theta_{\text{LoRA}}^{\text{encoder}}\right), \quad (5)$$

**LLM Decoder.** The representation of the encoder $\mathbf{H}_e$ is then fed to the LoRA-based LLM decoder to model the probability distribution of the linearized labeled sentence, denoted by $x$. Specifically, at the $i$-th step, the probability distribution of the output token $p(x_i)$ is calculated based on the encoded representation $\mathbf{H}_e$ and the previous decoder output $x_{<i}$ as follows:

$$\mathbf{h}_i = \text{LLM-Decoder}(\mathbf{H}_e; x_{<i}, \theta_{\text{LoRA}}^{\text{encoder}}) \quad (6)$$

$$p(x_i|x_{<i}) = \text{Softmax}(\mathbf{W}^\top \mathbf{h}_i + \mathbf{b}) \quad (7)$$

where $\mathbf{h}_i \in \mathbb{R}^d$ is the hidden representation of the $i$-th step, $\mathbf{W} \in \mathbb{R}^{d \times |\mathcal{V}|}$ and $\mathbf{b} \in \mathbb{R}^{|\mathcal{V}|}$ are learnable parameters, and $|\mathcal{V}|$ denotes the whole vocabulary size.

During the training stage, the parameters are optimized by minimizing the cross-entropy loss based on the teacher forcing method as follows:

$$\mathcal{L}^T = -\frac{1}{NM} \sum_{j=1}^{N} \sum_{i=1}^{M} \sum_{k=1}^{|\mathcal{V}|} t_{ik}^j \log p(x_{ik}^j), \quad (8)$$

where $N$ denotes the total number of samples, $M$ indicates the length of the linearized labeled sentence, $|\mathcal{V}|$ is the size of vocabulary, and $t_i^j$ refers to the ground-truth label distribution of the $i$-th word in the linearized labeled sentence of the $j$-th sample.

*3.3.3 Labeled Sentence Generation.* After training the Label-aware Multimodal Large Language Model, we utilize it to generate synthetic labeled sentences.

As shown in Figure 2, given the image and entity labels of a training sample, we feed its corresponding label-aware instruction and visual representations into the multimodal encoder of the trained LMLLM. For the LLM decoder, its initial input token is the  token representing the beginning of the decoded sentence, and the subsequent tokens are generated in an autoregressive manner based on a probability-based sampling strategy.

When sampling the next token, we use top-k [13] and top-p (nucleus) [17] sampling strategies to generate a synthetic labeled sentence. Firstly, the top-k method is utilized to retain the *top-k* tokens with the highest probabilities, and the sampling space is denoted as follows:

$$\mathcal{V}_i^{(k)} = \arg \max_{S \subseteq \mathcal{V}} \sum_{x \in S} (p(x|x_{<i})) \quad (9)$$

where $\mathcal{V}_i^{(k)}$ represents the vocabulary set comprising $k$ candidate tokens at time step $i$. Subsequently, within this subset of tokens, the top-p method is employed to retain those tokens with cumulative probabilities reaching the top-p threshold $pp$:

$$\mathcal{V}_i = \arg \min_{S \subseteq \mathcal{V}_i^{(k)}} |S| \quad \text{s.t.} \sum_{x \in S} (p(x|x_{<i})) \geq pp \quad (10)$$

where $\mathcal{V}_i$ represents the vocabulary set at time step $i$, encompassing all possible tokens that could occur in the sequence. Let $pp' = \sum_{x \in \mathcal{V}_i} p(x_i|x_{<i})$. The original distribution is re-scaled to a new distribution as follows:

$$p'(x_i|x_{<i}) = \begin{cases} p(x_i|x_{<i})/pp' & \text{if } x_i \in \mathcal{V}_i \\ 0 & \text{otherwise} \end{cases} \quad (11)$$

As the candidate tokens in $\mathcal{V}_i$ are predicted with higher probabilities, the generated sentence typically exhibits fluency and maintains proximity to the original training sample. Furthermore, owing to the inherent randomness in the sampling process, GMDA can sample different tokens as the next token, which enriches the diversity in the generated text.

The above process of token generation will be stopped when the next token is predicted as . After decoding a linearized labeled sentence, we can extract its entities $y^g$ based on the embedded labels within the sentence and obtain the synthetic sentence $s^g$.

## 3.4 Multimodal Image Generation

In this stage, our goal is to generate a corresponding image for each synthetic sentence. As the synthetic sentence is semantically related to the original image, we employ both the synthetic sentence and the original image as inputs for synthetic image generation, which can guide the model to refer to the original image during generation.

Specifically, given the synthetic sentence $s^g$ and its original image $v$, we utilize Stable Diffusion [28], a popular latent diffusion model, to generate a corresponding synthetic image $v^g$. Firstly, given an original image $v \in \mathbb{R}^{H \times W \times 3}$ in the RGB space, we feed it into the encoder of the variational autoencoder (VAE) in the Stable Diffusion model to obtain the latent representation:

$$x = \mathcal{E}(v) \quad (12)$$

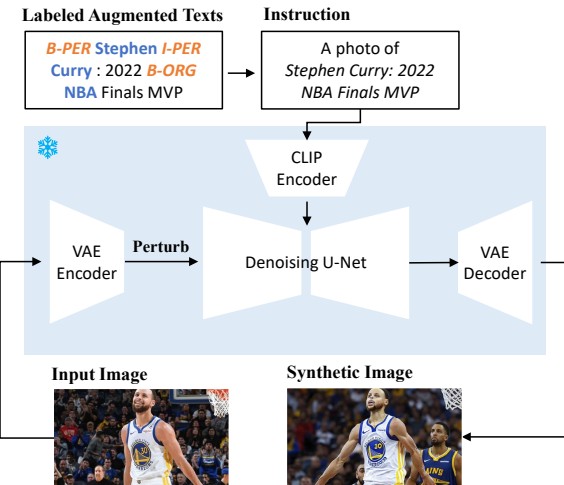

**Figure 3:** Overview of the Multimodal Image Generation stage.

where $\mathcal{E}$ refers to the VAE encoder and $\boldsymbol{x} \in \mathbb{R}^{h \times w \times c}$. At a specific time $t_0 \in (0, 1)$, the Gaussian noise with a standard deviation denoted as $\sigma^2(t_0)$ is added to $\boldsymbol{x}$, resulting in a perturbed latent representation $\boldsymbol{x_t}$ for denoising.

Secondly, given the synthetic sentence $\boldsymbol{s}^g$, we design a task instruction containing $\boldsymbol{s}^g$ as "*A photo of $\boldsymbol{s}^g$*", which is used to guide the generation of the synthetic image. Next, a CLIP text encoder [27] is utilized to project the task instruction to an intermediate representation, which is then mapped to the intermediate layers of the conditional denoising autoencoder. With the conditional denoising autoencoder, we can infer the denoised latent variant: $\tilde{x} \in \mathbb{R}^{h \times w \times c}$ and employ the decoder of the VAE model $\mathcal{D}$ to generate the synthetic image:

$$\boldsymbol{v}^g = \mathcal{D}(\tilde{x}) \tag{13}$$

where $\boldsymbol{v}^g \in \mathbb{R}^{H \times W \times 3}$ denotes the synthetic image.

It is worth noting that for the GMNER task, it is required to annotate the bounding box of each entity. In our preliminary experiments, we observe that the synthetic image is generally similar to the original image, and thus propose to replicate the ground-truth bounding boxes in the original image to cover the corresponding regions in the synthetic image, which are regarded as the bounding box annotation of each entity in the synthetic sentence.

### 3.5 Synthetic Data Filtering

To improve the quality of the synthetic labeled text-image pairs, we apply the following post-processing steps for data filtering: 1) We remove the text-image pairs whose text has less than 5 words. 2) We use the original training set $\mathcal{D}$ to train a base MNER or GMNER model and employ it to make predictions on the synthetic text-image pairs. If the predicted labels from the base model are inconsistent with the labels in the synthetic data, these text-image pairs will be removed. 3) We further remove the redundant text-image pairs with identical token and label sequences.

After data filtering, we can obtain a set of synthetic labeled data $\mathcal{D}_g = \left\{ \left( \boldsymbol{s}_i^g, \boldsymbol{v}_i^g, \boldsymbol{y}_i^g \right) \right\}_{i=1}^K$, which is then combined with the original training set for model training in each downstream task.

**Table 1:** Basic statistics of the three benchmark datasets

| Entity Type | Twitter-15 | | | Twitter-GMNER | | |
|---|---|---|---|---|---|---|
| | Train | Dev | Test | Train | Dev | Test |
| Person | 2,217 | 552 | 1,816 | 5,019 | 1,072 | 1,104 |
| Location | 2,091 | 522 | 1,697 | 1,918 | 407 | 404 |
| Organization | 928 | 247 | 839 | 3,035 | 595 | 638 |
| Miscellaneous | 940 | 225 | 726 | 1,807 | 376 | 397 |
| Total | 6,176 | 1,546 | 5,078 | 11,779 | 2,450 | 2,543 |
| # Tweet | 4,000 | 1,000 | 3,257 | 7,000 | 1,500 | 1,500 |

| Split | Twitter-FMNERG | | | | |
|---|---|---|---|---|---|
| | # Tweet | # Entity | # Entity Type | # Groundable | # Box |
| Train | 7,000 | 11,779 | 51 | 4,733 | 5,723 |
| Dev | 1,500 | 2,450 | 51 | 991 | 1,171 |
| Test | 1,500 | 2,543 | 51 | 1,046 | 1,254 |
| Total | 10,000 | 16,772 | 51 | 6,770 | 8,148 |

## 4 EXPERIMENTS

### 4.1 Experimental Settings

*4.1.1 Datasets.* To evaluate the effectiveness of the proposed GMDA framework, we conduct experiments on three publicly available Twitter datasets, i.e., Twitter-15 [41] for the MNER task and Twitter-GMNER [39] and Twitter-FMNERG [30] for the GMNER task. The basic statistics of each dataset are presented in Table 1. Note that Twitter-FMNERG is an extension of the Twitter-GMNER dataset, which extends the four coarse-grained entity types to 51 fine-grained entity types.

In addition to evaluating the GMDA framework on the standard full-supervision setting, we also construct three low-resource settings by randomly sampling 10%, 20%, and 40% data from the full training and development sets as the training and development sets. The whole test set is kept for model evaluation.

*4.1.2 Implementation Details.* For the Label-aware Multimodal Large Language Model (LMLLM), we employ the pre-trained InstructBLIP model released by Dai et al. [10], in which we adopt FlanT5-XL(3B) [9] as the LLM and ViT-g/14 [14] as the image encoder. For the LoRA [18] adapter in LMLLM, we adopt a rank of 8 and a dropout rate of 0.1. The batch size and the learning rate are set to 2 and 5e-5, respectively. For Multimodal Image Generation, we utilize the pre-trained Stable Diffusion v1.5 model [28], with a strength and guidance scale set to 0.8 and 10, respectively. All the models are implemented with PyTorch, and the Adam optimizer is adopted in the training stage. We run all the experiments on an NVIDIA RTX 3090 GPU.

*4.1.3 Evaluation Metrics.* Following previous studies [39, 41], we use Precision (Pre.), Recall (Rec.), and F1 Score (F1) as the evaluation metric for both MNER and GMNER tasks. The formula for computing the F1 Score is presented below:

$$F1 = \frac{2 \times Pre. \times Rec.}{Pre. + Rec.}, \tag{14}$$

where *Pre.* refers to the proportion of correctly predicted entity tuples among all predicted entity tuples, and *Rec.* denotes the proportion of correctly predicted entity tuples among all the ground-truth entity tuples. Note that for entity and type predictions, the predictions are regarded as correct only if they exactly match the ground-truth labels. For the object prediction in GMNER, if the

**Table 2:** Performance comparison between mixGen and the proposed GMDA framework on both MNER and GMNER tasks in low-resource settings.

| Task | Dataset | Methods | 10% | | | 20% | | | 40% | | |
|------|---------|---------|-----|-----|-----|-----|-----|-----|-----|-----|-----|
| | | | Pre. | Rec. | F1 | Pre. | Rec. | F1 | Pre. | Rec. | F1 |
| MNER | Twitter-15 | MMT5 | 63.91 | 63.86 | 63.88 | 68.70 | 70.90 | 69.79 | 73.16 | 75.38 | 74.25 |
| | | -w/ mixGen | 68.19 | 63.70 | 65.87 | 70.47 | 71.73 | 71.10 | 73.04 | 75.46 | 74.23 |
| | | -w/ GMDA | **69.59** | **67.31** | **68.43** | **72.94** | **73.00** | **72.97** | **73.87** | **76.38** | **75.10** |
| | | PGIM | 72.58 | 71.74 | 72.15 | 73.16 | 77.09 | 75.07 | 75.15 | 78.32 | 76.70 |
| | | -w/ mixGen | 71.40 | 74.12 | 72.74 | 75.39 | **77.20** | **76.29** | 76.74 | 77.90 | 77.32 |
| | | -w/ GMDA | **73.15** | **74.57** | **73.85** | **75.85** | 76.51 | 76.18 | **76.78** | **78.43** | **77.60** |
| GMNER | Twitter-GMNER | H-Index | 47.54 | 47.39 | 47.46 | 50.90 | 52.10 | 51.49 | 53.08 | 54.53 | 53.80 |
| | | -w/ mixGen | 45.72 | 50.84 | 48.15 | 50.70 | **52.65** | 51.66 | 53.75 | 53.44 | 53.59 |
| | | -w/ GMDA | **48.99** | **49.47** | **49.23** | **52.99** | 51.90 | **52.44** | **54.35** | **55.20** | **54.77** |
| | | TIGER | 47.84 | 51.66 | 49.67 | 50.54 | 55.14 | 52.74 | 52.29 | 57.03 | 54.56 |
| | | -w/ mixGen | 45.72 | 50.84 | 48.15 | 48.79 | 49.38 | 49.08 | 50.42 | 53.64 | 51.98 |
| | | -w/ GMDA | **49.28** | **53.12** | **51.13** | **52.00** | **56.01** | **53.93** | **54.81** | **57.75** | **56.24** |
| | Twitter-FMNERG | H-Index | 36.33 | 37.94 | 37.11 | 40.1 | 42.45 | 41.24 | 44.77 | 43.86 | 44.31 |
| | | -w/ mixGen | 38.46 | 39.11 | 38.79 | 41.36 | 42.64 | 41.99 | 43.00 | 44.57 | 43.77 |
| | | -w/ GMDA | **38.52** | **39.74** | **39.12** | **41.58** | **42.72** | **42.14** | **44.84** | **45.00** | **44.92** |
| | | TIGER | 34.77 | 36.96 | 35.83 | 41.2 | 43.48 | 42.31 | 41.96 | 46.02 | 43.89 |
| | | -w/ mixGen | 36.26 | 39.80 | 37.95 | 40.07 | 42.58 | 41.29 | 42.39 | 45.58 | 43.93 |
| | | -w/ GMDA | **38.31** | **41.00** | **39.61** | **41.26** | **44.81** | **42.96** | **43.71** | **46.32** | **44.97** |

object is groundable, we regard the prediction as correct when the maximum IoU score between the predicted object and all the ground truth bounding boxes exceeds 0.5; otherwise, if the object is not groundable, the prediction is correct only if it is None.

## 4.2 Comparison Systems

*4.2.1 Data Augmentation Baselines.* To evaluate the effectiveness of our GMDA framework, we consider two data augmentation methods for comparison: 1) Mix Generation (mixGen) [16] is a multimodal data augmentation method, which generates new image-text pairs by linearly interpolating images and concatenating text sequences from two existing image-text pairs. 2) Easy Data Augmentation (EDA) [34] is a text-only data augmentation method, which randomly performs one of the following operations on the original sentences: synonym replacement, random insertion, random swap, and random deletion. Since EDA can solely generate the synthetic sentence without incorporating a synthetic image, we pair the synthetic sentences with their corresponding original images to obtain synthetic text-image pairs.

*4.2.2 MNER and GMNER Baselines.* To show the effectiveness of the synthetic labeled data generated by GMDA, we adopt several competitive methods for MNER and GMNER as the base models: 1) *PGIM* [21] is the current state-of-the-art method on the MNER task, which leverages ChatGPT as an implicit knowledge base to generate auxiliary knowledge to enhance the performance of MNER. 2) *MMT5* [30] formulates the MNER task as a paraphrase generation task and employs a pre-trained Seq2Seq model VL-T5 [8] to generate the entity-type pairs based on the textual and visual inputs. 3) *H-Index* [39] is a hierarchical index generation framework for GMNER, which generates the entity-type-region triplets in a hierarchical manner with a pre-trained Seq2Seq model BART. 4) *TIGER* [30] is a T5-based multimodal Generation framework for GMNER, which directly generates the paraphrased target sequence containing entity-type-region triples from an image-text input pair.

In addition to the aforementioned base models, we further consider a number of representative MNER and GMNER methods for

comparison in the full-supervision setting: 1) UMT [38] is a unified Transformer framework for MNER, which captures the intermodal interactions. 2) UMGF [40] is a unified multi-modal graph fusion approach for MNER. 3) MNER-QG [19] is an end-to-end MRC-based MNER method with query grounding. 4) R-GCN [42] is a relation-enhanced Graph convolutional network for MNER. 5) CAT-MNER [33] is a Transformer-based MNER framework, which refines the cross-modal attention with expanding entity label words. 6) ICL-MNER [3] explores the potential of the in-context learning paradigm for few-shot MNER. 6) GVATT-RCNN-EVG [23], UMT-VinVL-EVG [38], UMGF-VinVL-EVG [40] and ITA-VinVL-EVG [32] are sequence labeling-based multimodal approaches for GMNER, which stack the EVG model over existing MNER methods introduced by Yu et al. [39].

## 4.3 Results in Low-Resource Settings

In Table 2, we compare the results of two multimodal data augmentation methods, i.e., mixGen and the proposed GMDA model in different low-resource settings.

*4.3.1 Results on MNER.* Based on the first six rows of Table 2, we can observe that the two multimodal data augmentation methods can generally bring improvements to the performance of the corresponding base models, especially in the extremely low-resource setting, i.e., only with 10% training data. Secondly, by comparing mixGen and GMDA, it is clear that using the augmented data generated from GMDA generally performs better. For example, in the 10% setting, when employing MMT5 as the base model, GMDA outperforms mixGen by 2.56 absolute percentage points in the F1 score. Lastly, for our GMDA framework, we can find that its performance improvement over PGIM is much smaller than that over MMT5. This is because the PGIM model leverages ChatGPT to generate auxiliary knowledge, which can perform much better than MMT5 in low-resource scenarios. Nevertheless, GMDA still improves the performance of PGIM by 1.7, 1.05, and 0.9 percentage points in 10%, 20%, and 40% settings, respectively.

**Table 3:** Results of different MNER methods in full-supervision settings.

| Methods | Twitter-15 | | | | | | |
| | Single Type(F1) | | | | Overall | | |
| | PER | LOC | ORG | OTH | Pre. | Rec. | F1 |
|---|---|---|---|---|---|---|---|
| UMT [38] | 85.24 | 81.58 | 63.03 | 39.45 | 71.67 | 75.23 | 73.41 |
| UMGF [40] | 84.26 | 83.17 | 62.45 | 42.42 | 74.49 | 75.21 | 74.85 |
| MNER-QG [19] | 85.68 | 81.42 | 63.62 | 41.53 | 77.76 | 72.31 | 74.94 |
| R-GCN[42] | 86.36 | 82.08 | 60.78 | 41.56 | 73.95 | 76.18 | 75.00 |
| CAT-MNER [33] | 88.04 | 84.70 | 68.04 | 52.33 | 78.75 | 78.69 | 78.72 |
| ICL-MNER [3] | - | - | - | - | 51.24 | 67.20 | 58.14 |
| MMT5 [32] | 86.37 | 83.32 | 66.00 | 46.52 | 74.57 | 78.13 | 76.31 |
|   -w/ EDA | 86.08 | 82.28 | 63.90 | **50.15** | 75.56 | 77.20 | 76.37 |
|   -w/ mixGen | 85.93 | 83.02 | 65.99 | 46.66 | 74.48 | 77.68 | 76.05 |
|   **-w/ GMDA** | **87.26** | **83.38** | **67.73** | 49.08 | **75.82** | **78.89** | **77.33** |
| PGIM [21] | 88.04 | 84.19 | 69.58 | 52.88 | 77.28 | 80.22 | 78.72 |
|   -w/ EDA | 88.13 | 84.09 | **70.32** | 52.03 | 76.69 | **80.97** | 78.78 |
|   -w/ mixGen | 87.73 | 84.00 | 68.45 | 51.50 | 77.17 | 78.90 | 78.03 |
|   **-w/ GMDA** | **88.39** | **84.35** | 69.77 | **53.70** | **78.32** | 80.07 | **79.19** |

*4.3.2 Results on GMNER.* As shown in the last 12 rows of Table 2, the comparison results on the GMNER task show similar trends to those on the MNER task. Specifically, for both H-Index and TIGER, using the data generated by GMDA brings consistent improvements in the F1 score in all the low-resource settings. For both Twitter-GMNER and Twitter-FMNERG datasets, the most significant improvements occur in the 10% setting, with an average F1 improvement of approximately 2% across different base models. Moreover, we find that in some cases, mixGen hardly yields any improvement. One possible reason is that linearly interpolating images and concatenating text sequences may disrupt the data distribution and thus introduce slight noise. In comparison to mixGen, our GMDA framework shows a more significant and consistent improvement, indicating that the augmented data generated from GMDA can well complement the original data.

These observations demonstrate the efficacy of the proposed data augmentation framework in low-resource settings.

## 4.4 Results in Full-Supervision Settings

In Table 3 and Table 4, we further compare the results of different models in the full-supervision setting.

Firstly, we can see from Table 3 that the base model MMT5 performs much better than most baseline MNER methods except CAT-MNER, while the other base model PGIM achieves the best performance among all the baseline methods. Secondly, when using synthetic data generated by mixGen, the F1 score even decreases, indicating that its data augmentation strategy may bring much noisy data. Lastly, it is easy to find that EDA and GMDA generally bring further improvements to the performance of MMT5 and PGIM. Concretely, EDA only obtains a very minor improvement (i.e., 0.06%) on both MMT5 and PGIM, while GMDA achieves 1.02 and 0.47 percentage points improvement in F1 score based on MMT5 and PGIM, respectively.

Similar to the performance trend in the MNER task, we can observe from Table 4 that the proposed GMDA framework consistently enhances the performance of H-Index and TIGER on both Twitter-GMNER and Twitter-FMNERG datasets. In contrast, it is clear that in the full-supervision setting, the two data augmentation baseline methods, i.e., EDA and mixGen, even lead to a slight

**Table 4:** Results of different GMNER methods in full-supervision settings.

| Methods | Twitter-GMNER | | | Twitter-FMNERG | | |
| | Pre. | Rec. | F1 | Pre. | Rec. | F1 |
|---|---|---|---|---|---|---|
| GVATT-RCNN-EVG [23] | 49.36 | 47.80 | 48.57 | 42.02 | 38.75 | 40.32 |
| UMT-VinVL-EVG [38] | 50.15 | 52.52 | 51.31 | 40.67 | 41.99 | 41.32 |
| UMGF-VinVL-EVG [40] | 51.62 | 51.72 | 51.67 | 41.73 | 42.11 | 41.92 |
| ITA-VinVL-EVG [32] | 52.37 | 50.77 | 51.56 | 43.05 | 42.51 | 42.78 |
| H-Index [39] | 56.16 | 56.67 | 56.41 | 46.83 | 46.28 | 46.55 |
|   -w/ EDA | 55.31 | 55.78 | 55.78 | 46.74 | **46.92** | 46.83 |
|   -w/ mixGen | **56.46** | 55.91 | 56.18 | 46.60 | 46.49 | 46.54 |
|   **-w/ GMDA** | 56.27 | **57.44** | **56.85** | **47.29** | 46.61 | **46.95** |
| TIGER [30] | 55.52 | 59.58 | 57.48 | **47.57** | 46.85 | **47.20** |
|   -w/ EDA | 57.06 | 57.48 | 57.27 | 45.10 | 48.08 | 46.54 |
|   -w/ mixGen | 55.53 | 59.40 | 57.40 | 44.86 | 48.18 | 46.46 |
|   **-w/ GMDA** | **57.09** | **60.21** | **58.61** | 45.58 | **49.30** | 47.37 |

**Table 5:** Ablation Study of H-Index with GMDA on the Twitter-GMNER dataset in the 10% low-resource setting.

| Methods | Pre. | Rec. | F1 |
|---|---|---|---|
| H-Index with GMDA | **48.99** | **49.47** | **49.23** |
|  - w/o Multimodal Text Generation | 47.56 | 47.90 | 47.73 |
|  - w/o Multimodal Image Generation | 49.36 | 48.61 | 48.98 |
|   training only Q-Former | 48.41 | 47.82 | 48.11 |

performance drop on the GMNER task, probably because their augmented data introduces excessive noise. This further demonstrates the advantage of GMDA over existing data augmentation methods.

## 4.5 In-Depth Analysis

*4.5.1 Ablation study.* To investigate the effectiveness of each component in the proposed GMDA framework, we choose to conduct an ablation study of GMDA on the Twitter-GMNER dataset in a 10% low-resource setting by using H-Index as the base model. Specifically, we compare the full GMDA model with its three ablations: 1) removing the Multimodal Text Generation stage; 2) removing the Multimodal Image Generation stage; 3) following the common practice to only fine-tune the Q-Former without adding LoRA adapters in the LMLLM.

As shown in Table 5, we can see that all the components in GMDA play an indispensable role in the overall performance. Firstly, removing the Multimodal Text Generation stage will significantly drop the performance, which shows that text data augmentation can increase the diversity of the sentences and is essential to GMDA. Secondly, discarding the Multimodal Image Generation stage also leads to a performance drop, which indicates that generating the synthetic image for each synthetic sentence can reduce the noise that arises from inconsistencies between the synthetic text and the original image. Lastly, only fine-tuning the Q-Former of the LMLLM also leads to a decrease in performance. This indicates the necessity of incorporating LoRA adapters for parameter-efficient fine-tuning.

*4.5.2 Case Study.* To better understand the GMDA framework, we conduct a qualitative analysis of two synthetic examples generated from GMDA. For comparison, we show the synthetic data generated from mixGen and GMDA in Table 6.

For the Original Sample A, since mixGen essentially concatenates it with another sample mentioning building entities, the synthetic

**Table 6:** Comparison between synthetic samples generated from mixGen and those generated from GMDA.

| Original Sample A | Synthetic Sample A | | Original Sample B | Synthetic Sample B | |
|---|---|---|---|---|---|
| | mixGen | GMDA | | mixGen | GMDA |
|  RT @tennis _ photos : Welcome back to No . 2 , **[Roger Federer, PER]** . http://t.co/OxQEOR7mKh |  RT @tennis _ photos : Welcome back to No . 2 , **[Roger Federer, PER]** . http://t.co/OxQEOR7mKh The Best way to visit the **[Louvre Museum, LOC]** . http://t.co/EqF2L87pw5 **[#paris, LOC]** http://t.co/CVdJN9BTEi |  RT @ESPN : Con-gratulations to @NRL_Hoopsplayer, **[Roger Federer, PER]** on winning #13 final! #BeLiveTNT http://t.co/G7bo5efVJL |  Oh I am just so excited ! # **[London-BurlesqueFestival, OTHER]** |  RT @MailChimp: **[Slate, PER]** wrote about the wonderful murals @Mon-Campana curated for our new office: Oh I am just so excited! # **[Lon-donBurlesqueFestival, OTHER]** |  @ **[LondonBurles-queTheater, OTHER]** The # **[London-BurlesqueFestival, OTHER]** will be an exciting week full of music and dancing! |

**Table 7:** Comparison results on the generated synthetic sentences in GMDA and those in EDA and mixGen.

| Criterion | Methods | Twitter-15 | Twitter-GMNER | Twitter-FMNERG |
|---|---|---|---|---|
| Similarity | EDA | 0.9283 | 0.9112 | 0.9109 |
| | MixGen | 0.6893 | 0.5998 | 0.6010 |
| | GMDA | **0.2898** | **0.2158** | **0.2156** |
| Diversity | EDA | 0.6838 | 0.6616 | 0.6608 |
| | MixGen | 0.5540 | 0.5436 | 0.5433 |
| | GMDA | **0.8915** | **0.8303** | **0.8399** |
| Perplexity | Origin | 133.76 | 569.23 | 569.23 |
| | EDA | 194.15 | 1070.28 | 1014.21 |
| | MixGen | 114.12 | **420.73** | **419.65** |
| | GMDA | **104.96** | 466.79 | 522.78 |

image tends to be noisy, containing unclear persons and buildings. In contrast, since our GMDA method revolves around the entity mentioned in the original sample, its synthetic sentence not only mentions the original entity, but also contains a different context, and the synthetic image is also quite relevant to the synthetic sentence.

Similarly, for the Original Sample B, the mixture sample generated from mixGen contains both PER entities and OTHER entities, which makes the mixed image unclear and may introduce noise to the original data. By contrast, the sample generated from GMDA focuses on keeping the original entity in the generated sentence and preserving the important visual regions in the original image. Moreover, the GMDA method also generates a new entity *London-BurlesqueTheater* and assigns a OTHER label to the entity, which shows the diversity of the synthetic data generated by GMDA.

*4.5.3 Analysis on Synthetic Sentences.* To evaluate the quality of sentences generated by GMDA, we conduct additional experiments in full-supervision settings and report the results in Table 7.

**Diversity.** Generating diverse contexts for entities can generally improve the model's robustness in entity recognition. To show the diversity of the generated data, we propose to measure the cosine similarity between the synthetic sentence and the original sentence, and calculate a Diversity score [20] denoting the percentage of unique n-grams in all the synthetic sentences. It can be observed from the first six rows of Table 7 that for EDA and mixGen, the synthetic sentences are generally similar to the original sentences and the percentage of unique n-grams is relatively low, while our GMDA method can generate diverse synthetic sentences with many unique n-grams, mainly due to the probability-based sampling strategy.

**Perplexity.** To evaluate the coherence of synthetic sentences, we further calculate the perplexity[1] of data generated from each compared method based on a pre-trained language model GPT-2 with $e$ as the base of the exponential function. In the last four rows of Table 7, we can observe that the perplexity of our GMDA framework is generally close to that of mixGen and the original sentences, while significantly lower than that of the EDA method. This shows that performing word replacement or insertion may disrupt the fluency of the original sentence and lead to high perplexity. In contrast, the Label-aware Multimodal Large Language Model in our GMDA method can help generate coherent sentences.

These observations demonstrate the superiority of the proposed GMDA framework over existing data augmentation methods regarding the diversity and fluency of the generated sentences.

## 5 CONCLUSION

In this paper, we proposed a Generative Multimodal Data Augmentation framework named GMDA to simultaneously generate synthetic text-image pairs and their corresponding high-quality entity annotations. GMDA contains two stages, i.e., a Multimodal Text Generation stage to generate synthetic labeled sentences with a Label-aware Multimodal Large Language Model (LMLLM) and a Multimodal Image Generation stage to generate the corresponding image for each synthetic sentence. Experiments on three benchmark datasets show that our GMDA framework consistently boosts the performance of several competitive methods for two subtasks of MNER in both standard and low-resource settings. Further analysis demonstrates the advantage of GMDA over existing data augmentation methods in terms of data diversity and fluency.

---

[1]https://huggingface.co/docs/transformers/perplexity

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
