# OpenReview forum: "Generative Multimodal Data Augmentation for Low-Resource Multimodal Named Entity Recognition"
_acmmm.org/ACMMM/2024/Conference — MM2024 Oral_

### Official Review · Reviewer_puX2 · 2024-05-05

**Rating:** 4
**Confidence:** 4

**Summary:**

The authors proposed a new data augmentation method with Multimodal Text Generation and Multimodal Image Generation for the task of MNER and GMNER. Extensive experiments show the proposed framework is effective.

**Strengths:**

The proposed data augmentation framework is well-motivated. The authors address the challenges properly. The paper is well-writen and easy to follow. The proposed framework could bring some insights for other researchers in the same filed.

**Limitations:**

The improvements under some specific settings (like in the 40% setting in Table 2) are not that much.

I have one important clarification question. In Section 3.5 (Line 512), the authors use the OORIGINAL training dataset D for filtering the synthetic data. Does that mean under the setting of 20%, you instead use the whole real training dataset? If so, this research cannot be regarded as under the LOW-RESOURCE setting because of the data leak. The authors should compare the proposed method with baselines which are trained using the whole training datatset. The final score would be changed regarding the answer of this question.

In addition, how many data are generated based on one real image-text pair? And how many data are left after filter? Does the proposed method sensetive to the number of generated data used? I hope there would be some empirical evidences regarding this.

**Suitability:**

3

---

### Official Review · Reviewer_WQC8 · 2024-05-10

**Rating:** 4
**Confidence:** 2

**Summary:**

The paper presents a novel framework called Generative Multimodal Data Augmentation (GMDA) for addressing the challenge of insufficient fine-grained annotated data in low-resource scenarios for Multimodal Named Entity Recognition (MNER). MNER is a crucial task in extracting structured information from multimodal content, such as text and images, found on social media platforms. The GMDA framework comprises two stages: Multimodal Text Generation and Multimodal Image Generation. It uses a Label-aware Multimodal Large Language Model (LMLLM) to generate labeled sequences based on prompts and associated images, followed by a Stable Diffusion model to generate semantically related synthetic images. The paper demonstrates the effectiveness of GMDA through experiments on three benchmark datasets, showing performance improvements for MNER in both full-supervision and low-resource settings.

**Strengths:**

1. The approach proposed in this paper has brought certain improvements over previous data augmentation strategies on many baseline methods, demonstrating the effectiveness of the two-stage data generation process.

2. The writing is fluent and there are no parts that are difficult to understand.

**Limitations:**

1. The proposed scheme in this paper appears to require a significant consumption of additional resources, such as training resources for Large Language Models (LLMs) and inference resources for Stable Diffusion. Moreover, the authors also suggest the need to train a base Multimodal Named Entity Recognition (MNER) model to serve as a classifier for synthetic images, which seems to make the proposed solution overly complex. Compared to methods like MixGen that do not require additional resource consumption, the costs associated with this approach in terms of various aspects need further clarification.

2. The paper mentions that when the original image contains object box annotations, it will 'replicate the ground-truth bounding boxes in the original image to cover the corresponding regions in the synthetic image.' Does this operation imply that the corresponding areas in the synthetic image are refilled by the corresponding areas from the original image? If so, this could lead to visual discontinuities at the filled boundaries. For instance, in Figure 3, the original bounding box for Curry does not fully cover Curry's portrait position in the generated image, which could result in the repetition of some features of the person, such as the right arm.

3. In the framework illustrated in Figure 2 of the paper, the output of the Large Language Model (LLM) seems to differ little from the original text input. If the entity types and positions are known in the original text (which is the case in the training set), it is entirely possible to generate the corresponding target text using some heuristic rules. However, this is obviously not feasible in the test set. So, I would like to ask the authors whether they expect the LLM to learn to automatically perform Named Entity Recognition (NER) on the input text? If that's the case, then in the context of MNER, it could be possible to have the LLM complete this task without the need for a two-stage data augmentation approach to enhance the richness of the images.

**Suitability:**

3

---

### Official Review · Reviewer_y8PV · 2024-05-22

**Rating:** 4
**Confidence:** 2

**Summary:**

This paper proposes a novel Generative Multimodal Data Augmentation (GMDA) framework for Multimodal Named Entity Recognition (MNER) to address the challenge of insufficient fine-grained annotated data, particularly in low-resource scenarios. Experimental results on three benchmark datasets show that GMDA effectively enhances the performance of several competitive MNER methods in both full-supervision and low-resource settings.

**Strengths:**

- The paper introduces a novel Generative Multimodal Data Augmentation (GMDA) framework, addressing the challenge of generating synthetic text-image pairs with high-quality entity annotations. This is a significant advancement in the field of Multimodal Named Entity Recognition (MNER) and Grounded Multimodal Named Entity Recognition (GMNER).
- Detailed comparisons with existing data augmentation methods and various MNER and GMNER baselines provide strong evidence of GMDA’s superior performance.
- The paper is clearly written, with a detailed explanation of the proposed GMDA framework, including its components and the experimental setup.

**Limitations:**

- Using the teacher forcing method during the training of the Label-aware Multimodal Large Language Model might result in lower quality generated sentences. Should strategies like Beam Search or Curriculum Learning be used to improve this?
- The ablation study on the effectiveness of each module should be conducted under diverse settings. Currently, it is only performed for H-Index with GMDA on the Twitter-GMNER dataset in the 10% low-resource setting.

**Suitability:**

3

---

### Official Review · Reviewer_thWV · 2024-05-24

**Rating:** 4
**Confidence:** 3

**Summary:**

This work addresses the challenge of insufficient fine-grained annotated data in Multimodal Named Entity Recognition (MNER), particularly in low-resource scenarios. The authors propose a Generative Multimodal Data Augmentation (GMDA) framework that includes two stages: Multimodal Text Generation and Multimodal Image Generation. Initially, annotated sentences are converted into labeled sequences, and a Label-aware Multimodal Large Language Model (LMLLM) is trained to generate these sequences using label-aware prompts and associated images. Then, a Stable Diffusion model generates synthetic images related to the generated sentences. Experimental results on benchmark datasets show that the GMDA framework significantly improves the performance of MNER methods in both full-supervision and low-resource settings.

**Strengths:**

1. The experiments were comprehensive.
2. Good results were achieved on all three datasets.

**Limitations:**

1. The authors did not provide source code for reproducibility.
2. There is a lack of deeper analysis and explanation in comparison with other methods to highlight the advantages of this approach.
3. The authors did not use a large model with a large number of parameters. Therefore, did they consider the effects of full fine-tuning or the impact of different PEFT methods on the model's performance before ultimately choosing LoRA?
4. In Table 3, are the results highlighted in bold all SOTA results?

**Suitability:**

2

---

### Meta-Review · Area_Chair_LNTC · 2024-06-30

**Recommendation:** Accept (Oral)
**Confidence:** 5

**Metareview:**

The paper presents Generative Multimodal Data Augmentation to address the challenge of insufficient fine-grained annotated data in low-resource Multimodal Named Entity Recognition.
Comparisons with existing data augmentation methods demonstrate the effectiveness of the proposed text and image generation,
showing superior performance. The paper is well-written and the proposed method offers insightful thoughts to other researchers in the field. All reviewers lean towards accepting this paper.